MINIREVIEW

# More is Different: Metabolic Modeling of Diverse Microbial Communities

Christian Diener,[a] Sean M. Gibbons[a,b,c]

[a]Institute for Systems Biology, Seattle, Washington, USA
[b]Departments of Bioengineering and Genome Sciences, University of Washington, Seattle, Washington, USA
[c]eScience Institute, University of Washington, Seattle, Washington, USA

**ABSTRACT** Microbial consortia drive essential processes, ranging from nitrogen fixation in soils to providing metabolic breakdown products to animal hosts. However, it is challenging to translate the composition of microbial consortia into their emergent functional capacities. Community-scale metabolic models hold the potential to simulate the outputs of complex microbial communities in a given environmental context, but there is currently no consensus for what the fitness function of an entire community should look like in the presence of ecological interactions and whether community-wide growth operates close to a maximum. Transitioning from single-taxon genome-scale metabolic models to multitaxon models implies a growth cone without a well-specified growth rate solution for individual taxa. Here, we argue that dynamic approaches naturally overcome these limitations, but they come at the cost of being computationally expensive. Furthermore, we show how two nondynamic, steady-state approaches approximate dynamic trajectories and pick ecologically relevant solutions from the community growth cone with improved computational scalability.

**KEYWORDS** flux balance analysis, metabolic modeling, metabolism, microbial communities, microbial ecology, systems biology

**M**icrobes are the second-largest contributor to biomass on Earth, and they carry out essential metabolic processes, ranging from nitrogen fixation in soil to nutrient absorption in the mammalian gut (1–3). Microbial phenotypes are strongly dependent on the uptake and secretion of metabolites, resulting in complex interaction networks with the surrounding biotic and abiotic components of the environment (4). Bacteria harbor a vast repertoire of enzymes that carry out millions of metabolic reactions (5). While one can infer the presence of individual enzymes from isolated genomes or metagenomically assembled genomes (MAGs), the realized metabolic phenotype of an organism is a complex function of its genetic capacity and its environmental context, including all of the ecological interactions with other organisms in the system.

Metabolic fluxes, or the rates of mass conversion by individual metabolic reactions (usually measured in units of mmol/[gDW · h]), are important and computationally convenient quantities for measuring the metabolic phenotype of an organism. The full set of metabolic fluxes for a given organism or community of organisms (i.e., the "fluxome") represents the flow of mass through the metabolic network (6). This fluxome provides a quantitative estimate of the metabolites that are consumed and produced by either a single organism or a set of organisms in a given environment. These estimates are extremely useful for biotechnological applications, such as the targeted production of biomolecules (7).

Metabolic fluxes can be measured directly via the isotopic labeling of molecular moieties (8). In addition, certain endpoint metabolites can also be measured longitudinally by using mass spectrometry to constrain their flux estimates (9). However, these experimental measurements can be expensive and low-throughput. While one can measure the

Address correspondence to Christian Diener, cdiener@isbscience.org, or Sean M. Gibbons, sgibbons@isbscience.org.

The authors declare no conflict of interest.

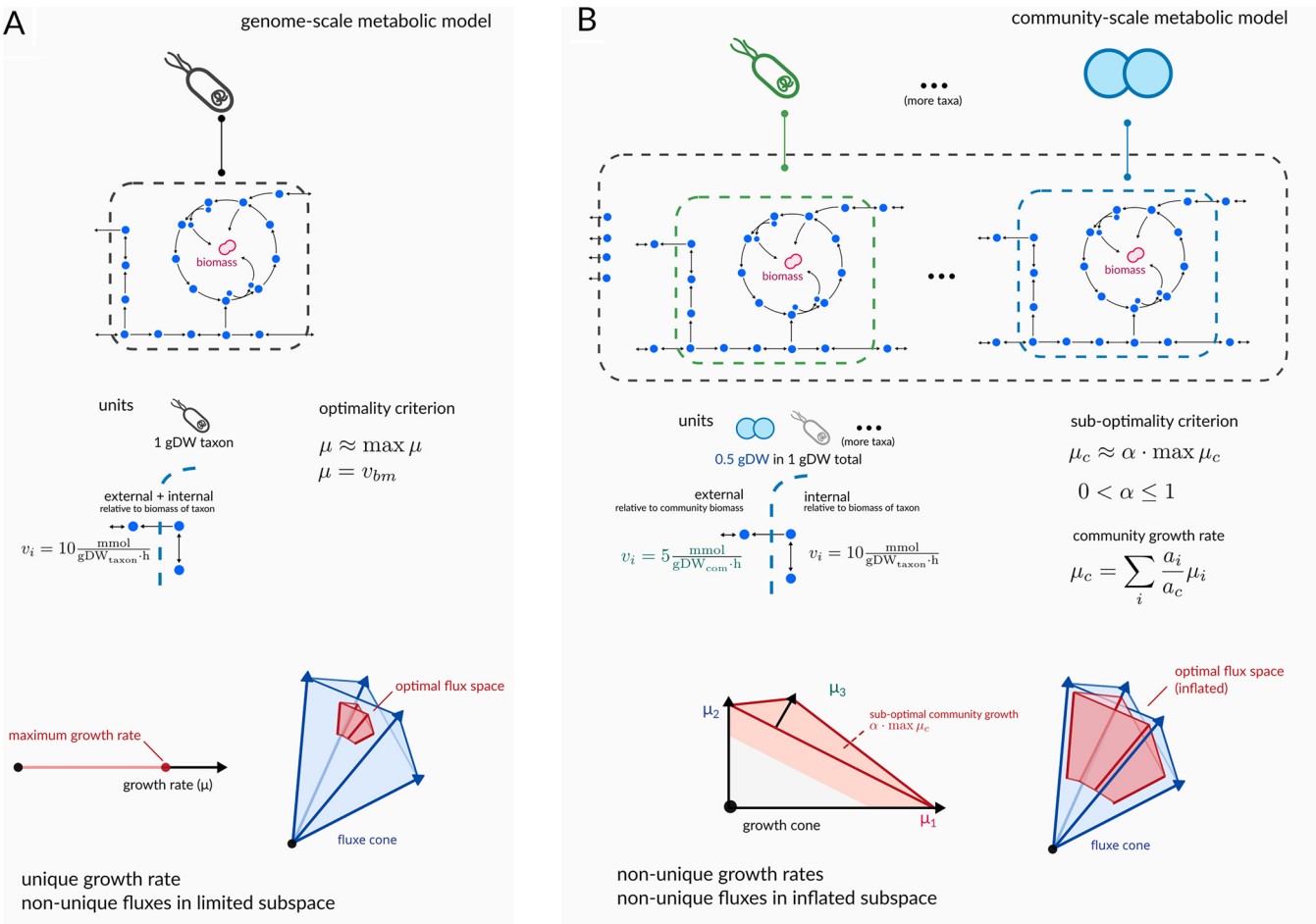

**FIG 1** Comparison between single taxon and community models. (A) In a single taxon genome-scale metabolic model, there is a unique optimal growth rate and an associated defined flux subspace. Fluxes that cross the model boundary represent exchanges with the external environment and are all relative to the biomass of the organism. The optimal growth rate solution for a single-taxon model can be represented as a point along a line and always has a unique maximum. (B) In a community model, individual taxon models are separate compartments that are embedded within a larger extracellular compartment. Transports across the taxon boundaries are scaled by that taxon's relative dry weight to maintain the mass balance and to fulfill the originally imposed flux bounds. Feasible growth rate solutions in a multitaxon community model form a polytope, and there is a suboptimal subspace containing achievable growth rate distributions. The feasible flux space for these community-scale models is greatly inflated, compared to single-taxon models.

total production or secretion rates for a single metabolite, it is much more difficult to map those rates to individual taxa within a complex community. Fortunately, metabolic fluxes can be predicted using a family of computational methods, called flux balance analysis (FBA) or constraint-based modeling, that integrate multiple layers of knowledge about the system and can be used to generate quantitative, mechanistically grounded hypotheses that are amenable to more direct experimental validation (10). FBA employs optimization techniques to infer a likely flux distribution from the set of metabolic reactions that is present in a genome and from external flux bounds, which are often derived from empirical data, that impose limits on uptake rates from the extracellular environment. These methods map an enzyme network to a set of irreversible metabolic reactions and their respective stoichiometries, thereby arranging them into a "stoichiometric matrix" ($S$) that describes all of the reactions that are present in the system (Fig. 1). The temporal change in metabolite abundances, $d\vec{x}/dt$, is then dictated by a metabolic flux vector, $\vec{v}(\vec{x})$, through the following ordinary differential equation:

$$d\vec{x}(t)/dt = S \cdot \vec{v}(\vec{x}).$$

Here, the flux vector $\vec{v}(\vec{x})$ captures the impact of enzyme properties, as well as enzyme and substrate abundance, whereas the stoichiometric matrix $S$ encodes the structure of the

metabolic network. Metabolic fluxes are driven by complex molecular kinetics, denoted $\vec{v}\left(\vec{x}, \vec{p}\right)$, with specific enzymatic constants $\vec{p}$ that are usually not known *a priori*. However, this can be circumvented by assuming steady-state conditions in which metabolite abundances are individually balanced between consumption and production and fluxes become constants, denoted $\vec{v}\left(\vec{x}\right) = \vec{v}$, which converts the resulting system of differential equations into a system of equalities that are commonly known as the balance equations:

$$S \cdot \vec{v} = 0.$$

Biomass production is usually dictated by its own metabolic reaction, with a normalized stoichiometry consuming 1gDW per mmol, resulting in units of 1gDW/(gDW · h), or 1/h (i.e., equivalent to the growth rate of the organism). Thus, the steady-state assumption implies a constant growth rate, which further implies exponential growth. Reactions are often separated into forward and reverse reactions to convert the problem into a standard form in which $\vec{v} \geq 0$. Additionally, the balance equations are usually paired with individual flux bounds ($v_i \leq b_i$) that dictate whether a reaction is reversible (by either allowing or prohibiting flux through the reverse reaction), impose a global maximum flux (e.g., due to a diffusion limit), or determine the availability of metabolites in the extracellular environment by setting bounds on import fluxes.

Fluxes are often obtained by maximizing the growth rate, assuming that optimal growth in a given environment is an evolutionarily realistic objective for a bacterium. This assumption is known to be imperfect, as it is often unclear which selection pressures have acted on an organism in the context of a particular environment (11). For example, a maximal growth rate may not be favored above metabolic flexibility and phenotypic plasticity in the context of a rapidly fluctuating environment (12–14). Nevertheless, near-maximal growth appears to be a realistic objective in most cases, as has been observed in both mechanistic modeling and *in vitro* evolution experiments (15, 16).

## METABOLIC MODELING OF MICROBIAL COMMUNITIES

Transitioning from a single-taxon metabolic model to a multitaxon model brings a set of unique challenges that go beyond the mere increase in complexity due to the larger numbers of metabolic reactions and metabolites. In microbial community-scale metabolic models, individual taxon-specific metabolic models are embedded in their own compartments within a large extracellular compartment that contains all of the taxa (17). Exchanges between taxa and the extracellular environment are mapped from the original transport reactions from the individual models, and additional exchanges with the extracellular compartment are added for all unique metabolites that can be exchanged across organisms. Some care must be taken to correctly scale the exchanges between the taxa and the extracellular compartment in order to maintain mass balance in the system. Fluxes and flux bounds are usually expressed relative to the dry weight of a given organism, which becomes less well-defined in a microbial community in which all taxa are present in different relative abundances and in which there is an absolute biomass of the entire microbial consortium. For instance, if a taxon can take up glucose with an upper flux bound of 10 mmol/(gDW · h) then 0.5 gDW of the organism can take up to 5 mmol of glucose per hour, which has to be accounted for to maintain mass balance (Fig. 1). This can be achieved either by scaling the flux bounds of each organism, according to its abundance, or by scaling the exchanges across the individual taxon boundaries (18, 19). In practice, we consider the second strategy to be more stable, as it avoids small flux bounds for low-abundance taxa that can lead to numerical issues. Furthermore, the second strategy puts all metabolic fluxes across all taxa on the same scale (i.e., per 1gDW per hour), which allows for the direct comparison of fluxes between taxa. These kinds of comparisons usually do not require knowledge of the total mass of the microbial consortium and of the individual taxa, but rather, knowledge about the relative contribution of each taxon to the total dry weight in the community. In practice, relative abundance data from metagenomic or amplicon sequencing can be used as a proxy for the relative dry weights. However, these sequencing-based estimates can be distorted due to differences in detection efficiencies, the total community biomass, and quantification

error (20). Thus, these proxies, while convenient and widely available, may introduce biases into the analysis.

An even larger challenge is posed by the optimization criterion. As discussed above, the assumption that fluxes follow a pattern that allows for the maximum growth and fitness of a single taxon in a given environmental context is already imperfect, and it is even less clear what constitutes maximum fitness in a microbial community. One possible measure of fitness (e.g., reproductive success) is the growth rate for the entire microbial community, denoted $\mu_c$, which can be derived from the sum of abundances, denoted $a_i$, for a set of $n$ taxa (21). Here, observing an exponential growth process for the sum of taxon abundances and using the summation rule of differentiation yields:

$$d/dt \sum_{i=1}^{n} a_i(t) = \sum_{i=1}^{n} a_i(t) \cdot \mu_i = a_c \sum_{i=1}^{n} \frac{a_i}{a_c} \mu_i = a_c \cdot \mu_c,$$

where $a_c$ denotes the total abundance of the community, which is itself dictated by an exponential growth process with an intrinsic growth rate, namely, $\mu_c$, which is given by the sum of the individual growth rates scaled by the relative dry weight of each organism:

$$\mu_c = \sum_{i=1}^{n} \frac{a_i}{a_c} \mu_i.$$

Thus, while there may be many weighting schemes for combining the individual growth rates into a single objective, only weighting by relative abundance is consistent with the optimization of the growth rate of the microbial community. Despite its tractability, maximizing the overall community growth rate $\mu_c$ instead of the individual taxon growth rates $\mu_i$ has several limitations. First, whereas an individual organism may be evolutionarily optimized to achieve a growth rate in proximity to its maximum, there is no evolutionary justification for why unrelated microbial taxa would be driven to maximize overall community biomass. This theoretical maximal community growth rate may, for instance, require a less abundant taxon to sacrifice its own growth in order to provide nutrients to a more abundant taxon, which would run counter to our understanding of evolution. Indeed, microbial consortia do not act as a single organism, and individual taxon growth rates are almost certainly a result of taxon-specific biomass maximization in the presence of complex intertaxon interactions that are dominated by competition for metabolic resources (22, 23). Previous experimental work with *in vitro* microbial communities has shown that the average, community-wide growth rate and biomass decline with increasing species richness (down to as little as approximately 60% of the achievable maximum), likely due to the costs imposed by interspecies competition (24). However, stable microbial consortia still show reasonably fast community-scale growth rates, implying that there exists a trade-off, denoted $\alpha$, ranging from zero to one (likely closer to one than to zero), representing the fraction of the maximum community growth that is actually achieved in a given community, which lies in a suboptimal region that is proximal to the maximal community growth plane (Fig. 1B). Because $\mu_c$ is a linear combination of individual taxon growth rates, there can be an infinite number of unique sets of $\mu_i$ for any given $\mu_c$. In more technical terms, the maximum community growth forms a Pareto front (i.e., a hyperplane in multidimensional space), but the suboptimal community growth forms a subspace (Fig. 1B). Suboptimal community growth and uncertainty in taxon growth rates within this subspace propagate into uncertainty in the overall flux space, thereby inflating the set of feasible flux solutions and making it difficult to extract biologically meaningful results from metabolic modeling (Fig. 1B). Most linear programming solvers tend to only report a single solution, with a bias toward sparse solutions (i.e., feasible solutions with zero growth rates for the majority of taxa) that can be constructed from simpler solution bases. In this context, merely maximizing the community growth rate often yields solutions in which a single abundant taxon grows at its maximum theoretical rate and all other taxa support the growth of this focal taxon, thereby resulting in improbably high growth rates for the abundant taxon and evolutionarily

unrealistic ecological interactions between unrelated taxa (i.e., one species sacrificing its growth for another) (25).

If we wish to remain within this biomass optimization paradigm, which works well in single-species models, the question becomes how we might leverage ecological and evolutionary principles to further constrain these multispecies systems and generate more realistic predictions.

## BUILDING MORE ACCURATE MICROBIAL COMMUNITY MODELS

The simple optimization of the overall community growth rate in multispecies metabolic models does not result in biologically realistic growth rates for individual taxa (25). Prior experimental work suggests that competition between microbes can result in a total biomass production rate that is smaller than its theoretical maximum (24). Consequently, the real growth rate distribution is likely located somewhere in an area close to, but not necessarily coinciding with, the optimal plane (Fig. 2). In order to select meaningful growth rate distributions, additional principles need to be integrated into the modeling framework.

Microbial communities necessarily pass through distinct growth phases as they approach a steady-state (Fig. 2A). Thus, a steady-state solution must be accessible from an initial inoculation state that corresponds to the origin of the growth cone in which none of the taxa are growing. Dynamic FBA methods have become popular approaches by which to simulate this full trajectory from a set of initial conditions, to the steady-state, and then a subsequent return to the stationary phase if the system lacks a constant supply of nutrients (Fig. 2B) (26–29). Alternative dynamical approaches also include individual-based dynamic FBA models, in which single organisms or cells are modeled on a two-dimensional or three-dimensional grid with a discrete time-stepping scheme, continually updating the concentrations of extracellular metabolites and the taxon abundances in time and space (30, 31). Dynamic FBA, as well as individual-based models, require knowledge of kinetic parameters for each uptake reaction in the system, which can range into the hundreds for single organism models and into the thousands for microbial consortia. One of the advantages of individual-based modeling is that they do not require an explicit community-level objective. The community growth rate emerges as a consequence of the growth maximization of individual taxa in a shared environment, which is much closer to the ecological processes that we suspect are taking place in complex microbial communities. Individual-based models also require knowledge about initial conditions, such as the inoculation abundances and the concentrations of metabolites, at the start of the trajectory. Steady-state abundances, growth rates, and fluxes are then emergent properties of these models. A major downside to these dynamical models is that they can be lengthy to simulate for many taxa or for longer time frames. Furthermore, large fluxes in the models will require small time steps so as to not overshoot the consumption of metabolites in the extracellular environment, resulting in hundreds to thousands of steps within a single trajectory. Thus, these dynamical approaches may be well-suited to simulate fewer than a dozen or so taxa in reasonable time frames, but they become computationally intractable when simulating many samples from real-world microbial ecosystems that contain dozens to hundreds of interacting taxa (32).

Although dynamical models can be computationally infeasible for diverse, real-world microbial systems, there are approaches for approximating solutions that lie close to those that are obtained by simulating the full growth trajectory (Fig. 2C and D). Growth rates usually increase monotonically and nonlinearly from the inoculation point until they reach the community-specific optimal hyperplane. Thus, a simple approach is to choose the optimal points in the growth rate hyperplane that are most likely to be reached by such a trajectory. In particular, one could ask for a simple linear approximation to the trajectory, interpreted as a line from the origin of the growth cone to a particular growth rate solution. Although not specifically designed with trajectories in mind, two previously developed steady state optimization strategies, namely, SteadyCom and cooperative trade-off FBA (ctFBA), turn out to approximate trajectories. SteadyCom (18), one of the first methods to use this kind of approximation, assumes a perfectly diagonal trajectory through the growth

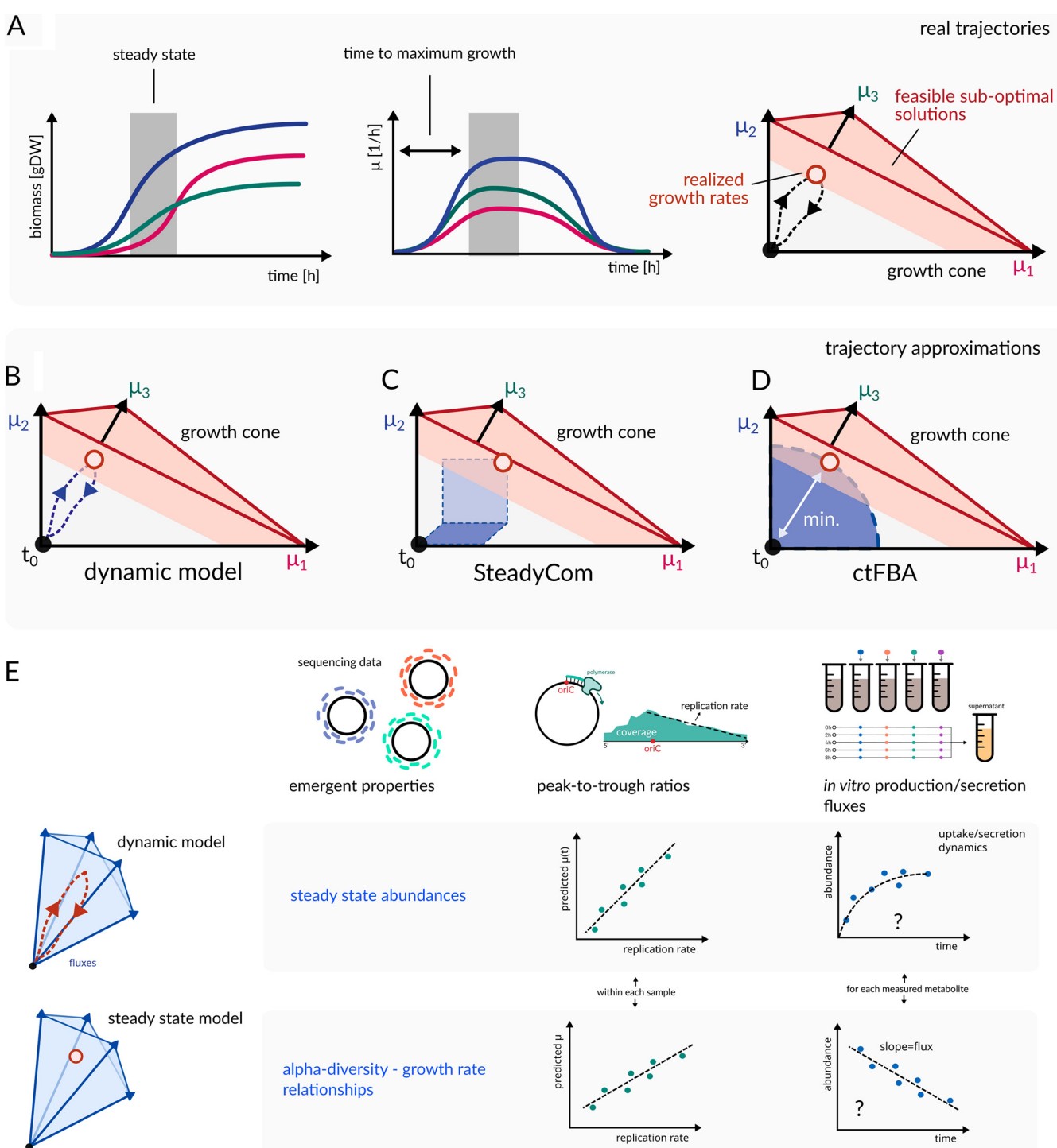

**FIG 2** Approaches to selecting optimal solutions from the community growth cone. Growth rates in complex microbial communities are often dictated by a trajectory from the inoculation time to the point of maximum growth (red circle) located in a suboptimal growth region (red shaded area) (A). A dynamic model will simulate the trajectory explicitly (B), but one may also try to directly find a point in the growth cone that is empirically close to the true, unknown trajectory. SteadyCom uses a perfect diagonal as the implicit trajectory (C), whereas ctFBA finds the shortest path between the inoculation point and the maximum growth regime (red circle) (D). Untargeted approaches for the large-scale validation of community-scale metabolic model predictions (E). Emergent properties, such as the predicted versus observed taxon abundances that are estimated from amplicon or shotgun sequencing or the relationships between community diversity and growth rates, can be validated. Peak-to-trough ratios from deep metagenomic shotgun sequencing or isolate sequencing can be used to estimate replication rates (proxies for growth rates), which can be compared to model-predicted growth rates. Supernatants from *in vitro* culturing that are sampled at various time points can be used to measure community-wide consumption and production fluxes with untargeted metabolomics, which are then compared to model-predicted fluxes.

cone in which all growth rates are equal (Fig. 2C). This trajectory implies: (i) that taxa grow independently of one another and (ii) that the steady state itself is maintained by a linear dilution/death progression that acts the same on all taxa. Based on these assumptions, all growth rates end up taking on the same value. SteadyCom can be implemented in a community-scale metabolic model as a set of additional constraints that is followed by a maximization of the characteristic growth rate. In the original SteadyCom implementation, this rate maximization was also used to generate the relative abundance distribution of all taxa in the sample by finding the relative abundances that yielded the largest characteristic growth rate. However, this is not *per se* required, and SteadyCom can be employed even with a known relative abundance distribution. This procedure results in a unique optimal growth rate distribution and only requires a single solution step of a linear programming problem, which scales well to diverse communities. Because all taxa need to grow at the same rate in SteadyCom (18), the characteristic growth rate is highly sensitive to the smallest rate of growth in the model, making the approach brittle when a single taxon cannot grow in a particular environment due to an incomplete description of the available nutrients or missing biochemical reactions in the metabolic model. Furthermore, we have widespread empirical evidence that taxa in naturally occurring microbial communities show different growth rates, likely due to intrinsic factors (e.g., rRNA copy numbers), competition, or complex dilution and death processes that affect taxa differently, which clearly violate the assumptions of SteadyCom.

Based on the computational intractability of dynamical methods and the apparent empirical violations of the assumptions of methods such as SteadyCom, we previously suggested an alternative nondynamical approach for approximating growth rates, called cooperative trade-off FBA (ctFBA), that does not require homogeneous dilution/death processes or growth independence. Motivated by the requirement that most abundant taxa observed in a sample should be able to grow, ctFBA identifies the growth rate distribution that is closest to the inoculation point in the growth cone (i.e., the shortest path to the empirical abundance distribution from the inoculation point) (Fig. 2D) (25). Thus, the ctFBA implies: (i) that microbial communities will try to reach their optimal growth phases as quickly as possible and (ii) that during the initial phases of growth, environmental resources are plentiful, allowing different taxa to grow in parallel in this low-biomass, nutrient-rich environment. In order to account for the loss of theoretically maximal community biomass production, as described above, this method also incorporates a trade-off parameter, denoted $\alpha$, that defines the percentage of community growth that is lost due to competition or evolutionary tradeoffs. ctFBA requires the solution of a linear programming problem that is followed by a quadratic programming problem, which is less computationally efficient than SteadyCom but is much more efficient than dynamic FBA models and can be applied to hundreds of taxa and thousands of samples (25). Additionally, ctFBA is amenable to the inclusion of empirically quantified growth rates for a subset of organisms. Specifically, one can fix a subset of growth rates to known values and deploy ctFBA for only those taxa with unknown growth rates. Overall, ctFBA strikes a balance between computational tractability and biologically reasonable assumptions.

All of the methods discussed above, whether they are dynamic or linear approximations, are capable of identifying a unique set of growth rates, which greatly limits the possible flux space. An overview of available methods can be found in (33). It should be noted that while these models do not reduce the flux space to a single solution, they do overcome the "curse of dimensionality" that is introduced by transitioning from single taxon to multitaxon microbial community models. With a unique set of growth rates in place, one can introduce further constraints on the flux space to obtain a low variability solution via approaches such as parsimonious FBA, which chooses the solution with the lowest enzyme use requirement, or the minimization of the community-wide import flux rates which chooses the flux distributions with the lowest overall resource use requirements (25, 34).

## TOWARDS PREDICTIVE MECHANISTIC MODELS OF MICROBIAL COMMUNITIES

All of the community-scale metabolic modeling strategies discussed above will ultimately require experimental validation to assess their utility and translational potential.

Metabolic models predict many quantities that can be compared to experimental measures. The gold standard methods for measuring intracellular and extracellular fluxes are isotope labeling experiments (8, 35, 36). However, model predictions can be validated via various approaches (Fig. 2E). At the highest level, one can compare the emergent properties of different modeling approaches to sequencing data from real world ecosystems. For example, one can compare the predicted steady-state abundances for the taxa from dynamic models with taxon abundances that were measured using amplicon or metagenomic shotgun sequencing. This strategy is less useful for steady-state models, in which amplicon or metagenomic data are often used as constraints in the model building. However, steady-state models also have emergent properties that can be compared to empirical data. For instance, SteadyCom will force growth rates to be similar across taxa, and these can be directly compared to empirically-derived growth rates that were obtained from optical density (OD) measurement time courses or from metagenome-inferred replication rates (37). Similarly, growth rate predictions from dynamic models and from ctFBA can be validated using empirical growth or replication rates (25). Another emergent property of ctFBA that could be tested empirically is an implied relationship between alpha diversity and growth rates, in which especially low-abundance taxa show higher growth rates in more diverse microbial communities (see [25] for a derivation). Finally, one can also run large-scale, direct experimental validation of quantitative flux predictions via longitudinal metabolomics sampling from microbial communities *in vitro* or *ex vivo* (9). In particular, empirical measures of community-wide metabolite production and consumption rates are excellent targets for model validation, as these rates are easier to obtain than species-specific fluxes (Fig. 2E).

Through more widespread empirical validation, we will gain a better understanding of where each modeling approach succeeds or fails, which will ultimately pave the way toward quantitative community-scale predictions. Ultimately, more accurate and more computationally-tractable community-scale metabolic modeling approaches have the potential to revolutionize bioengineering, industrial manufacturing, personalized nutrition, precision health care, sustainable agriculture, and environmental conservation.

## ACKNOWLEDGMENTS

This research was funded by the Washington Research Foundation Distinguished Investigator Award and by startup funds from the Institute for Systems Biology (to S.M.G.). S.M.G. is also supported by the National Institute of Diabetes and Digestive and Kidney Diseases of the National Institutes of Health (NIH) under award no. R01DK133468.

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
