## [Reviewer comments · mSystems]

More is different: metabolic modeling of diverse microbial communities

Christian Diener and Sean Gibbons

Corresponding Author(s): Sean Gibbons, Institute for Systems Biology

Review Timeline:

Submission Date:	December 16, 2022
Editorial Decision:	January 8, 2023
Revision Received:	February 25, 2023
Accepted:	February 27, 2023

Editor: Jack Gilbert

Reviewer(s): The reviewers have opted to remain anonymous.

Transaction Report:

DOI: <https://doi.org/10.1128/msystems.01270-22>

Prof. Sean M. Gibbons
Institute for Systems Biology
401 Terry Ave. N
Room 359
Seattle, WA 98109

Re: mSystems01270-22 (More is different: metabolic modeling of diverse microbial communities)

Dear Prof. Gibbons:

The reviewers were generally supportive but have comments that definitely require addressing in a resubmission.

Reviewer comments are found at the end of this letter.

Your minireview is likely to be accepted once the indicated changes are made.

Author Bios: If you would like a brief biographical sketch of each author (limit, 150 words) to be published at the end of your article, please submit text and photos with your modified manuscript. For complete guidelines on submission requirements, please see the journal Submission and Review Process requirements at <https://journals.asm.org/journal/mSystems/submission-review-process>. **Submissions of a paper that does not conform to mSystems guidelines will delay acceptance of your manuscript.**

Figures [**Editor: insert figure numbers here**] in your manuscript are good candidates for graphical enhancement. We now offer our authors the services of ASM's contracted artist, Patrick Lane of ScEYence Studios. This art enhancement service is free of charge to authors of minireviews and full-length reviews, and turnaround time is fast. Please contact Patrick on receiving this letter. Complete contact information for Patrick and further instructions are posted at <https://journals.asm.org/pb-assets/pdf-text-excel-files/graphical-enhancement-support.pdf>

Please return your modified manuscript within 60 days; if you cannot complete the modification within this time period, please contact me. If you decide that you do not want to modify the manuscript and wish to submit it to another journal, please notify me of your decision immediately so that the manuscript can be formally withdrawn.

To submit the modified manuscript, log onto the eJP submission site at <https://msystems.msubmit.net/cgi-bin/main.plex>. If you cannot remember your password, click the "Can't remember your password?" link and follow the instructions on the screen. Go to Author Tasks and click the appropriate manuscript title to begin the resubmission process. The information you entered when you first submitted the paper will be displayed. Please update the information as necessary. Provide (1) point-by-point responses to the issues raised by the reviewers as file type "Response to Reviewers," not in your cover letter, and (2) a PDF file that indicates the changes from the original submission (by highlighting or underlining the changes) as file type "Marked Up Manuscript - For Review Only."

To submit your modified manuscript, log onto the eJP submission site at <https://msystems.msubmit.net/cgi-bin/main.plex>. If you cannot remember your password, click the "Can't remember your password?" link and follow the instructions on the screen. Go to Author Tasks and click the appropriate manuscript title to begin the resubmission process (ONLY the corresponding author will have access to the full record for resubmission). The information that you entered when you first submitted the paper will be displayed. Please update the information as necessary and do the following:

- 1) Provide point-by-point responses to the issues raised by the reviewers in a file designated as "Response to Reviewers" (NOT the cover letter).
- 2) Upload ALL of your source files (not PDF and not just the files requiring modification) and make sure that all elements meet the technical requirements for production.
- 3) Do not provide a highlighted or tracked-changes copy of the paper in the main manuscript upload. This should be a clean copy instead. You may provide the compare copy separately by uploading it as a "Marked-Up Manuscript" file.
- 4) Make sure that the figure legends are included in the main manuscript file (not uploaded separately).

ASM policy requires that data be available to the public upon online posting of the article, so please verify all links to sequence records, if present, and make sure that each number retrieves the full record of the data. If a new accession number is not linked or a link is broken, provide production staff with the correct URL for the record. If the accession numbers for new data are not publicly accessible before the expected online posting of the article, publication of your article may be delayed; please contact

the ASM production staff immediately with the expected release date.

If you would like to submit an image for consideration as the Featured Image for an issue, please contact mSystems staff.

Sincerely,

Jack Gilbert
Editor, mSystems

Journals Department
Reviewer comments:

Reviewer #1 (Comments for the Author):

Diener and Gibbons provide a clear and well-written review of metabolic modeling, and the challenges when scaling it up to the community/metagenomic scale. I applaud the authors for succinct and approachable writing that is, independently of my comments below, a worthy and needed contribution to the literature in the field.

My main comment is about the need and focus for a perspective that is focused on ctFBA. ctFBA is described as a new approach (in L21 it's even stated "Here, we describe an approach", which is somewhat more befitting of a research paper than a perspective), but it seems to have in fact been published as part of MICOM back in 2020. Are there any differences between MICOM and ctFBA? If not, what is the role of this perspective, if the work has already been done? Have new challenges been identified, or new approaches to solve them?

Additionally--and just to clarify, the following is a suggestion, and not something I expect addressed--but I wonder if the authors have considered expanding on what they wrote to provide an objective review of methods in this realm. They're off to a really fantastic start - just addition of a few additional methods and a more objective reference to MICOM/ctFBA - and I feel they have a unique perspective and contribution that would greatly advance the field.

Some additional comments:

1. My impression of MICOM was that its assumptions are different to the ones listed in L245-248, more about realistic growth rates. Is that so? Are there differences?
2. L234-237 - requires citations.
3. The drawing in Fig. 2C/D was not intuitive for me. Can the authors offer intuition for the shapes in 2C and the circle in 2D?

Reviewer #2 (Comments for the Author):

This perspective article describes different stoichiometric constraint-based approaches for modeling microbial communities. This is a timely topic, given the rising interest in pairing metagenomic data-driven analyses of communities with mechanistic models, to help understand how communities assemble, and how their functions vary as a function of environmental parameters and taxonomic abundances.

The authors touch on interesting concepts, and strike a reasonable balance between not being too technical (for the purpose of appealing to a broad-audience), and providing some glimpse of how stoichiometric models really work. At the same time the article contains several poorly phrased and inaccurate statements, very sparse references, and oversimplified descriptions that gloss over major conceptual and technical challenges. Writing a new perspective is great and I think it is important for people to appreciate the challenges and opportunities of building ecosystem-level models, but I think that the manuscript need significant revision. In addition to the specific issues listed below, I would suggest that the authors try to sharpen their overall message, trying to focus more clearly on what new perspective they are bringing to the field.

A few points about the Abstract: (i) The abstract reads more like the abstract of a research article than of a review paper, where it sounds like some new ideas are being introduced, whereas, as far as I can tell, these are all previously published results. It is also awkward to see the proverbial "Here, we describe..." sentence after other sentences describing what the articles report. I think the abstract should clearly state what is being reviewed, and what new perspective the authors bring to the field. (ii) Other

people may have used this terminology before, but I find the term "Metagenome-scale metabolic models" quite ambiguous. My first instinct, when I hear metagenome-scale, is that this is about a modeling approach that uses metagenomic data, where enzymatic functions can be modeled from the data itself, even if knowledge about what organisms they belong to is lacking. What the authors discuss here is rather ecosystem-level models, where multiple organisms (potentially built from MAGs) are simulated together. I would personally recommend using something like "whole community scale", or at least define clearly in the text what they mean by metagenome-scale metabolic models. (iii) The authors write that "there is currently no consensus for what the fitness function of an entire community should look like in the presence of ecological interactions". I would argue that there isn't even a consensus on whether it is even possible and meaningful to define a fitness function for a community. This touches upon complex issues like group selection, that clearly cannot be described in the abstract, but I feel this statement already makes assumptions that are debatable.

Line 38: "Quantification of the metabolic phenotype can be done by measuring metabolic fluxes". I feel that this presents an overly narrow view of community-level metabolic phenotypes. Fluxes are hugely important and are a (computationally) convenient set of variables, but they are not the only ones, and we don't really know whether they are sufficient to quantify the metabolic phenotypes of a community. Also, there are many other approaches to quantify phenotypes of a community, e.g. quantification of growth rates of organisms, macromolecular composition, intracellular and extracellular metabolites.

Line 46: Flux measurement not only are low-throughput and expensive, but they are also typically only capturing whole system net fluxes. This issue comes back also at the end of the manuscript, but I think it is crucially important. People have measured a few fluxes for individual species in bioreactors, or overall net fluxes in a community. However, I think that it is extremely difficult, conceptually and technically, to measure fluxes between different species in a community, and - as far as I know - no one has really done this.

Line 49: "Alternatively, metabolic fluxes can be predicted using a family of computational methods, called flux balance analysis (FBA) or constraint-based modeling".

I think this statement, as phrased, is problematic. Models are very useful, but they cannot be considered an alternative to experimental measurements, especially given that researchers are just starting to test how these models work for new environmental isolates and communities. I assume this is not what the authors meant, so the sentence should be modified.

Line 57: "temporal change in metabolite abundances, $dx(t)/dt$, is then dictated by a metabolic flux vector $v(x)$ ". This sentence does a poor job at capturing what this equation is about. What dictates the dynamics are the enzyme abundances and properties and substrate concentrations (collectively encoded in the flux variables), and the network structure, encoded in the S matrix.

Line 60: This equation and several other equations in the text are not rigorously described. There is no mention of x being a vector, and it is not self-evident that dx/dt implies a component by component set of equations. If the goal is to avoid mathematical formalisms for the sake of broad-audience appeal, then you could find a way to describe concepts in words, but if equations are used, all variables should be well defined, including vector and matrix nomenclature.

Line 62: "Metabolic fluxes are often driven by complex molecular kinetics $v(x, p)$ ". Why only "often"? Are there cases when fluxes are not driven by kinetics?

Lines 72-73: Why only mention upper bounds to v ? Also, the notation v

Lines 79-88: Several years of research from multiple groups are summarized here without a single reference.

Line 88: "Fluxes are usually expressed relative to the dry weight of a given organism...". This is indeed how they are always expressed in traditional FBA for individual organisms. The question of how this translates to multi-compartment models for multiple organisms is an interesting one. Again, the authors should add references. For example, the statement "In practice, the second strategy is often preferred, as it avoids very small flux bounds for low-abundance taxa that can lead to numerical issues." Seems important. But often preferred by whom?

Lines 93-95: I must admit that I am a bit puzzled by how this whole argument is presented, in what I think of an upside-down (and potentially confusing) way... The argument made here is that due to the different abundances of species in a community, fluxes have to be somehow rescaled. This seems to gloss over the fact that one of the reasons people try to build community-level models in the first place is to predict community composition. Simple compartment-based models (unlike the approaches describe later in the manuscript) cannot do this, due to the steady state approximation (for the same reason intracellular metabolite concentrations cannot be predicted, I believe). I think the authors should revisit this part considering this. If abundances are an input to the model, what is the goal and desired output? If abundances are a desired output, how can this be achieved? As for the rest of these papers, statements should be backed by references (or explicitly expressed as authors' opinions).

Line 114 Equation: I have never seen before the inverted definition equal sign used at the end of the equation. I understand what the authors mean, but not everyone might.

Line 134: "First, while it is a well-established principle that an individual organism is evolutionarily optimized to achieve

a growth rate close to its maximum."

This is a reasonable assumption used by many, but I strongly disagree that this is a well-established principle. There is abundant literature exploring alternative objectives (PMID: 17625511), showing that the same organism may be close to optimal under some conditions but not other (PMID: 12415116), hinting to the importance of tradeoffs and Pareto frontiers (PMID: 22556256), and demonstrating how in some instances optimal growth rate is not the main evolutionary goal (PMID: 23818838).

Line 111: "One computationally-tractable approach would be to maximize the growth rate for the entire microbial community, μ_C , which can be derived from the sum of abundances, a_i , for a set of n taxa." and Line 132 and following paragraphs: "maximizing the overall community growth rate μ_C "

I have several questions about this approach:

(i) Many multi-compartment models (including what I think is the very first described in the literature, (PMID: 17353934) define ecosystem level objectives as individual community-growth reactions that take different biomasses and combine then with a fixed stoichiometry (like different biomass components in the biomass reaction of a single organism). This approach, used by many afterwards, is not described here. Maximizing the weighted sum of the biomasses of the different organisms is a very different approach.

(ii) The maximization of the weighted sum of biomasses, because of the nature of linear programming, is likely to slide the solution to maximize the biomass of the organisms that gives the largest numerical reward, with other species going to zero. The newer approaches described later by the authors get around this issue, but I think the problem is not just an issue with optimizers, as claimed, but a fundamental limitation of the approach (in the same way as one would not want to simulate growth of a single organism by maximizing the weighted sum of all biomass components).

Line 145: "Second, there are infinitely many sets of individual taxon growth rates μ_i for any given μ_C ." What is the evidence or reference for this statement? It is not self-evident, and I could imagine scenarios where a single optimum for the community growth is achieved with no alternative solution. Also, it is not clear to me whether the authors here refer to multiple alternative optima or to a real Pareto frontier, where tradeoff for multiple objectives exist. The two concepts are related but distinct, and this is not described clearly here.

Line 150: "Most linear programming solvers tend to only report a single solution, with a bias towards sparse solutions". As mentioned above (comment about Line 132) I think this is a deeper issue than just optimizers returning a single solution.

Line 168: "but not necessarily coinciding with, the optimal plane". Again, not clear what this optimal plane is and why we would expect it to exist. Is it a set of multiple alternative optima? Where has this been shown? I would expect more explanations and references.

Line 177: "Dynamic flux models are costly to simulate due to the proliferation of kinetic parameters for each biochemical reaction in the system, which can range into the thousands for single organism models and into the millions for microbial consortia."

This is not true. Dynamic FBA models do not require kinetic parameters for each biochemical reaction in the system. They only require kinetic parameters for uptake reactions. Also, while slightly more elaborate than regular FBA, dynamic FBA is basically as efficient as FBA, and the only issue of computational cost has to do with the number of steps required. Again, in addition to the inaccurate statement, there are no citations of the original Dynamic FBA work (PMID: 12202358), or of the now numerous studies that used it to simulate communities (e.g. PMID: 20665645, PMID: 24794435, PMID: 24742179, PMID: 24992662).

Line 179 and following lines: "individual-based" models is used in an ambiguous and inaccurate way. The authors here conflate real individual-based models (i.e. where individual cells are modeled, e.g. PMID: 28531184, PMID: 22817898) with dynamic models where multiple individual species are simulated in the same environment, but there is no attempt to model individual cells (PMID: 34635859, PMID: 24742179).

Line 194: "A major downside to these dynamical models is that they often do not scale well to many taxa or long time frames. Furthermore, large fluxes in the models will require very small-time steps so as not to overshoot consumption of metabolites in the extracellular environment, resulting in hundreds-to-thousands of steps within a single trajectory leading to lengthy simulations." Again, this statement is inaccurate. Dynamic FBA can handle hundreds of species, and the computing time scales linearly with the number of species. Scaling with time is also in general linear. Moreover, people have explored strategies for dynamic allocation of time steps. Also: it should be obvious, but simulations are long in proportion to the time course they simulate, which is unavoidable if one wants to learn about temporal variations of a community.

270: "All of the community-scale metabolic modeling strategies discussed in this perspective will ultimately require experimental validation to assess their usefulness and translational potential."

I agree. However, it is surprising that there is no mention at all of the several previous papers that compared model predictions with experiments, including references mentioned in comments about Line 177 and many more.

288: "Finally, we argue for the direct experimental validation of quantitative flux predictions either by isotopic labeling or by longitudinal metabolomics sampling from microbial communities (9)."

Who wouldn't argue in favor of this? The problem, as I hinted at the beginning, is that it is extraordinarily difficult to measure fluxes in a community (except for overall net fluxes in and out of the whole system). The few examples I know where measurement of fluxes across species has been attempted (for two species) is through extremely sophisticated machines like nanoSIMS (PMID: 27419912, PMID: 32788385). If the authors are aware of other promising technologies or efforts, they should mention them. So, overall I find this concluding statement quite naive, disconnected from reality (again no references, except a global metabolomics one) and with no hint to how this may be possible. At the same time, some of the other major limitations in the field of ecosystem-level modeling (including the challenges of building reliable models for newly sequenced organisms, the lack of information on biomass composition, metabolic composition of the environment, non-metabolic interactions, e.g. signaling, antibiotic killing etc.) are not mentioned at all.

Replies to reviewers

Reviewer #1

Diener and Gibbons provide a clear and well-written review of metabolic modeling, and the challenges when scaling it up to the community/metagenomic scale. I applaud the authors for succinct and approachable writing that is, independently of my comments below, a worthy and needed contribution to the literature in the field.

Response: We thank the reviewer for the positive feedback and great comments.

My main comment is about the need and focus for a perspective that is focused on ctFBA. ctFBA is described as a new approach (in L21 it's even stated "Here, we describe an approach", which is somewhat more befitting of a research paper than a perspective), but it seems to have in fact been published as part of MICOM back in 2020. Are there any differences between MICOM and ctFBA? If not, what is the role of this perspective, if the work has already been done? Have new challenges been identified, or new approaches to solve them?

Response: This is a good point, and we apologize for the confusion. The focus of our manuscript was more on general methods that can overcome the limitations of suboptimal community growth and alternative taxon growth rate distributions. We have now made this more clear in the abstract (lines 15-21) and in the main text.

Additionally--and just to clarify, the following is a suggestion, and not something I expect addressed--but I wonder if the authors have considered expanding on what they wrote to provide an objective review of methods in this realm. They're off to a really fantastic start - just addition of a few additional methods and a more objective reference to MICOM/ctFBA - and I feel they have a unique perspective and contribution that would greatly advance the field.

Response: Agreed, and again, apologies for any confusion. The manuscript was first submitted as a perspective piece but was reclassified to a minireview based on the length of the text. We started to create a supplemental table that lists a wider array of community-scale modeling implementations, but then we were made aware of a recent preprint by Scott Jr., Benito-Vaquerizo et al. (<https://doi.org/10.1101/2023.02.08.527721>) that provides a much more comprehensive overview, along with a stringent evaluation of model performances. Thus, we decided to cite this reference in the text (lines 246-247) and hope this provides the reader with a fuller picture of the different types of models that are available and their various capabilities.

Some additional comments:

1. My impression of MICOM was that its assumptions are different to the ones listed in L245-248, more about realistic growth rates. Is that so? Are there differences?

Response: That is true. The original motivation for ctFBA was to ensure non-zero growth rates for observed taxa in a sample, and we showed how this requires quadratic optimization. We also showed that this leads to more empirically accurate growth rate estimates, though we gave no intuitive reason why that should be the case in the original MICOM paper. In this manuscript, we now provide an intuitive explanation for why ctFBA belongs to a class of simple trajectory-approximating methods, how ctFBA is more computationally efficient/tractable than dynamic FBA when simulating diverse communities across hundreds-to-thousands of samples, and how ctFBA can be motivated from ecological principles.

2. L234-237 - requires citations.

Response: We added a citation to the original SteadyCom manuscript in line 216. There is no direct citation for the remainder of the statement but it is a direct consequence of the SteadyCom requirement that all growth rates are equal in the community which limits the growth rate of any taxon to the smallest maximum rate achievable by any other taxon.

3. The drawing in Fig. 2C/D was not intuitive for me. Can the authors offer intuition for the shapes in 2C and the circle in 2D?

Response: We provided some more explanation in the main text in lines 162-166 now and also specifically stress that the region and point denote what is annotated in panel B of the figure (lines 400-406).

Reviewer #2 (Comments for the Author):

This perspective article describes different stoichiometric constraint-based approaches for modeling microbial communities. This is a timely topic, given the rising interest in pairing metagenomic data-driven analyses of communities with mechanistic models, to help understand how communities assemble, and how their functions vary as a function of environmental parameters and taxonomic abundances.

The authors touch on interesting concepts, and strike a reasonable balance between not being too technical (for the purpose of appealing to a broad-audience), and providing some glimpse of how stoichiometric models really work. At the same time the article contains several poorly phrased and inaccurate statements, very sparse references, and oversimplified descriptions that gloss over major conceptual and technical challenges. Writing a new perspective is great and I think it is important for people to appreciate the challenges and opportunities of building ecosystem-level models, but I think that the manuscript need significant revision. In addition to the specific issues listed below, I would suggest that the authors try to sharpen their overall message, trying to focus more clearly on what new perspective they are bringing to the field.

Response:

We are glad that the reviewer agrees with us on the need for a discussion/perspective on challenges and possible solutions when it comes constructing community-scale models. We apologize for the lack of stringency in our definitions and a rather sparse coverage of the available literature in our original draft. Some of this was owed to (a) the intent to keep the manuscript succinct and easily-accessible to a wider audience and (b) space and citation number limitations in the article category we aimed for (we initially submitted this work as a perspective article, but the editor has decided to make it a mini-review, which allows us to add more content to the revision).

In this vein we appreciate the detailed comments of the reviewer, which have helped us to improve the manuscript and convey a clearer message. We have added major revisions to the manuscript and we have attempted to better articulate the goals of this mini-review. We believe these revisions have improved the quality and clarity of our work.

A few points about the Abstract: (i) The abstract reads more like the abstract of a research article than of a review paper, where it sounds like some new ideas are being introduced, whereas, as far as I can tell, these are all previously published results. It is also awkward to see the proverbial "Here, we describe..." sentence after other sentences describing what the articles report. I think the abstract should clearly state what is being reviewed, and what new perspective the authors bring to the field. (ii) Other people may have used this terminology before, but I find the term "Metagenome-scale metabolic models" quite ambiguous. My first instinct, when I hear metagenome-scale, is that this is about a modeling approach that uses metagenomic data, where enzymatic functions can be modeled from the data itself, even if knowledge about what organisms they belong to is lacking. What the authors discuss here is rather ecosystem-level models, where multiple organisms (potentially built from MAGs) are simulated together. I would personally recommend using something like "whole community scale", or at least define clearly in the text what they mean by metagenome-scale metabolic models. (iii) The authors write that "there is currently no consensus for what the fitness function of an entire community should look like in the presence of ecological interactions". I would argue that there isn't even a consensus on whether it is even possible and meaningful to define a fitness function for a community. This touches upon complex issues like group selection, that clearly cannot be described in the abstract, but I feel this statement already makes assumptions that are debatable.

Response:

(i) We agree. We have rewritten the last few sentences of the abstract (lines 15-21) to condense the message further and we have revised the main text to avoid any misinterpretation about this article being a mini-review.

(ii) This is a good point and we understand how this term might be misinterpreted. As the reviewer mentions, we used this term because it is somewhat established in the field, although we may have played some part in that ourselves. We have now renamed all instances of “metagenome-scale” to “community-scale”, which is close to what was recommended by the reviewer (dropping the word “whole”, because these models often do not include all microbial taxa in the community, like fungi or protists).

(iii) We completely agree with the reviewer that the contention that multi-species microbial communities are evolutionarily selected for maximum community growth has little evidence and is likely incorrect. This was our original point. However, many currently-employed community-scale models make this exact assumption. Unfortunately, we failed to convey that this very problem is exactly what is addressed by the methods mentioned in the manuscript, and we have now added additional explanation to the text (lines 127-135). We disagree with the reviewer in that it is not “even possible [...] to define a fitness function for a community”. One can certainly define a fitness function for a community (like we did in the manuscript) and growth rates, in general, are certainly valid measures of fitness or reproductive success. We do agree, however, that there are no coherent evolutionary explanations for how a community of unrelated species would evolve towards maximizing community-level growth. On the other hand, it is also clear that selection pressures on individual organisms push them close to their own maximal growth, and that these individual pressures ultimately propagate to community-level growth/biomass. Thus, while community-level growth is unlikely to be perfectly optimal, it is nevertheless likely to be only slightly suboptimal. We have now included a more detailed discussion of these points in the manuscript (lines 135-143).

Line 38: "Quantification of the metabolic phenotype can be done by measuring metabolic fluxes". I feel that this presents an overly narrow view of community-level metabolic phenotypes. Fluxes are hugely important and are a (computationally) convenient set of variables, but they are not the only ones, and we don't really know whether they are sufficient to quantify the metabolic phenotypes of a community. Also, there are many other approaches to quantify phenotypes of a community, e.g. quantification of growth rates of organisms, macromolecular composition, intracellular and extracellular metabolites.

Response: Agreed, and we have now revised this statement to be in more line with what the reviewer suggests (lines 33-35).

Line 46: Flux measurement not only are low-throughput and expensive, but they are also typically only capturing whole system net fluxes. This issue comes back also at the end of the manuscript, but I think it is crucially important. People have measured a few fluxes for individual species in bioreactors, or overall net fluxes in a community. However, I think that it is extremely difficult, conceptually and technically, to measure fluxes between different species in a community, and - as far as I know - no one has really done this.

Response: We did not mean to make any statements about measuring taxon-specific fluxes and have revised the text in lines 255-259 and 273-277. We have also revised Figure 2 to clarify

which fluxes we think are good targets for validation. We do believe that validation of community-wide secretion and uptake fluxes is experimentally tractable and can increase confidence in other flux predictions that would be more difficult to measure empirically.

Line 49: "Alternatively, metabolic fluxes can be predicted using a family of computational methods, called flux balance analysis (FBA) or constraint-based modeling".

I think this statement, as phrased, is problematic. Models are very useful, but they cannot be considered an alternative to experimental measurements, especially given that researchers are just starting to test how these models work for new environmental isolates and communities. I assume this is not what the authors meant, so the sentence should be modified.

Response: Agreed, we replaced the word "alternatively" on line 46.

Line 57: "temporal change in metabolite abundances, $dx(t)/dt$, is then dictated by a metabolic flux vector $v(x)$ ". This sentence does a poor job at capturing what this equation is about. What dictates the dynamics are the enzyme abundances and properties and substrate concentrations (collectively encoded in the flux variables), and the network structure, encoded in the S matrix.

Response: We have revised this statement as suggested by the reviewer (lines 58-60).

Line 60: This equation and several other equations in the text are not rigorously described. There is no mention of x being a vector, and it is not self-evident that dx/dt implies a component by component set of equations. If the goal is to avoid mathematical formalisms for the sake of broad-audience appeal, then you could find a way to describe concepts in words, but if equations are used, all variables should be well defined, including vector and matrix nomenclature.

Response: We have now revised the formulas in lines 57-58, 60, 64, 65-66, and 70 to use the common arrow vector notation, where appropriate. We mentioned repeatedly in the text that S is a 'matrix'.

Line 62: "Metabolic fluxes are often driven by complex molecular kinetics $v(x, p)$ ". Why only "often"? Are there cases when fluxes are not driven by kinetics?

Response: We have corrected this phrase on line 60.

Lines 72-73: Why only mention upper bounds to v ? Also, the notation $v < b$ should be defined. As mentioned above, these are vectors, and it is not self-evident that it should be taken component by component.

Response: For the sake of simpler visualization, we consider convex optimization problems to be in standard form. We have now made this clear in the text in lines 69-74 and we also mentioned that this requires splitting forward and reverse reactions.

Lines 79-88: Several years of research from multiple groups are summarized here without a single reference.

Response: We apologize that formatting constraints kept us from including a larger sampling of the literature, making our citations somewhat biased. We have now added additional key citations to lines 85-101.

Line 88: "Fluxes are usually expressed relative to the dry weight of a given organism...". This is indeed how they are always expressed in traditional FBA for individual organisms. The question of how this translates to multi-compartment models for multiple organisms is an interesting one. Again, the authors should add references. For example, the statement "In practice, the second strategy is often preferred, as it avoids very small flux bounds for low-abundance taxa that can lead to numerical issues." Seems important. But often preferred by whom?

Response: We have now made it clear that this is our own preference and we have included citations to other implementations that also use this same scaling (lines 99-103).

Lines 93-95: I must admit that I am a bit puzzled by how this whole argument is presented, in what I think of an upside-down (and potentially confusing) way... The argument made here is that due to the different abundances of species in a community, fluxes have to be somehow rescaled. This seems to gloss over the fact that one of the reasons people try to build community-level models in the first place is to predict community composition. Simple compartment-based models (unlike the approaches describe later in the manuscript) cannot do this, due to the steady state approximation (for the same reason intracellular metabolite concentrations cannot be predicted, I believe). I think the authors should revisit this part considering this. If abundances are an input to the model, what is the goal and desired output? If abundances are a desired output, how can this be achieved? As for the rest of these papers, statements should be backed by references (or explicitly expressed as authors' opinions).

Response: While one can use certain steady-state methods to estimate abundances (SteadyCom, for example), we disagree that this is the major motivation to build community-scale metabolic models. Indeed, our research group tends to be much more interested in estimating metabolic fluxes (e.g., short-chain-fatty-acid production by human gut communities in the context of a specific diet, which is highly relevant to human health). If one is specifically interested in modeling only taxon abundances, models like generalized Lotka-Volterra models, consumer-resource models, or logistic growth models are much simpler and more direct. For many complex microbial communities, measurements of which taxa are present also come with a built-in estimate of relative abundances (e.g., in the case of amplicon or shotgun DNA sequencing). Thus, having initial constraints on abundances and then wanting to estimate metabolic fluxes is a very common use-case for community-scale metabolic models. As such, we concentrate here on this particular use-case.

Line 114 Equation: I have never seen before the inverted definition equal sign used at the end of the equation. I understand what the authors mean, but not everyone might.

Response: This was not essential for the manuscript and we have removed it (see lines 120-121).

Line 134: "First, while it is a well-established principle that an individual organism is evolutionarily optimized to achieve a growth rate close to its maximum."

This is a reasonable assumption used by many, but I strongly disagree that this is a well-established principle. There is abundant literature exploring alternative objectives (PMID: 17625511), showing that the same organism may be close to optimal under some conditions but not other (PMID: 12415116), hinting to the importance of tradeoffs and Pareto frontiers (PMID: 22556256), and demonstrating how in some instances optimal growth rate is not the main evolutionary goal (PMID: 23818838).

Response: Agreed. We have now revised this section and added the references mentioned by the reviewer (lines 75-82).

Line 111: "One computationally-tractable approach would be to maximize the growth rate for the entire microbial community, μ_C , which can be derived from the sum of abundances, a_i , for a set of n taxa. " and Line 132 and following paragraphs: "maximizing the overall community growth rate μ_C "

I have several questions about this approach:

(i) Many multi-compartment models (including what I think is the very first described in the literature, (PMID: 17353934) define ecosystem level objectives as individual community-growth reactions that take different biomasses and combine them with a fixed stoichiometry (like different biomass components in the biomass reaction of a single organism). This approach, used by many afterwards, is not described here. Maximizing the weighted sum of the biomasses of the different organisms is a very different approach.

(ii) The maximization of the weighted sum of biomasses, because of the nature of linear programming, is likely to slide the solution to maximize the biomass of the organisms that gives the largest numerical reward, with other species going to zero. The newer approaches described later by the authors get around this issue, but I think the problem is not just an issue with optimizers, as claimed, but a fundamental limitation of the approach (in the same way as one would not want to simulate growth of a single organism by maximizing the weighted sum of all biomass components).

Response:

(i) We now mention this approach in the text, although we believe that it has some serious drawbacks. As mentioned in the manuscript in lines 124-126, while this is a common approach in multi-objective optimization, those objectives are 1) not optimizing a valid growth rate or 2) introducing implicit assumptions about the steady state abundances. We disagree that this is similar to choosing the coefficients in the biomass reaction because these coefficients are

experimentally-measured relative contributions to the community biomass and are thus the organism-equivalent of the relative abundance weighting scheme we and others are using (<https://doi.org/10.1371/journal.pone.0064567>).

(ii) This would be true if that particular case would indeed be the only solution yielding the maximum. As we have shown previously (<https://doi.org/10.1128/mSystems.00606-19>), there are usually many alternative growth rate distributions yielding maximum community growth in naturally-occurring communities. Allowing for suboptimal community growth, for instance 60% of the theoretical maximum as observed experimentally in <https://doi.org/10.1038/s41467-021-21844-7>, allows for even more flexibility in taxon growth rates. The model formulation itself does not favor one solution over the another, but the convex solver will. So we do believe this is ultimately also an issue of how solvers pick a solution. Nevertheless, we have revised the respective sections in the manuscript (lines 149-156).

Line 145: "Second, there are infinitely many sets of individual taxon growth rates μ_i for any given μ_C ." What is the evidence or reference for this statement? It is not self-evident, and I could imagine scenarios where a single optimum for the community growth is achieved with no alternative solution. Also, it is not clear to me whether the authors here refer to multiple alternative optima or to a real Pareto frontier, where tradeoff for multiple objectives exist. The two concepts are related but distinct, and this is not described clearly here.

Response: We have revised this section to make it clear that the optimization expression itself allows for infinitely many distinct solutions, but that the constraints can modulate this (lines 143-149).

Line 150: "Most linear programming solvers tend to only report a single solution, with a bias towards sparse solutions". As mentioned above (comment about Line 132) I think this is a deeper issue than just optimizers returning a single solution.

Response: As argued above, this would only be the case if that would indeed be the only optimal solution, which we do not believe is commonly the case. This is particularly true in the case of suboptimal community growth regimes, which we consider here.

Line 168: "but not necessarily coinciding with, the optimal plane". Again, not clear what this optimal plane is and why we would expect it to exist. Is it a set of multiple alternative optima? Where has this been shown? I would expect more explanations and references.

Response: We have now expanded this section in the manuscript to make this point clearer (lines 135-149).

Line 177: "Dynamic flux models are costly to simulate due to the proliferation of kinetic parameters for each biochemical reaction in the system, which can range into the thousands for single organism models and into the millions for microbial consortia."

This is not true. Dynamic FBA models do not require kinetic parameters for each biochemical reaction in the system. They only require kinetic parameters for uptake reactions. Also, while slightly more elaborate than regular FBA, dynamic FBA is basically as efficient as FBA, and the only issue of computational cost has to do with the number of steps required. Again, in addition to the inaccurate statement, there are no citations of the original Dynamic FBA work (PMID: 12202358), or of the now numerous studies that used it to simulate communities (e.g. PMID: 20665645 , PMID: 24794435 , PMID: 24742179, PMID: 24992662).

Response: We apologize for conflating dynamic kinetic models with dynamic FBA models, which do indeed require fewer parameters, though they still require on the order of hundreds of parameters (see revised lines 177-179). As the reviewer states, the longer computation times mostly stem from optimizing the model at every time step along the trajectory. We have made this clear in the text (lines 186-190).

Line 179 and following lines: "individual-based" models is used in an ambiguous and inaccurate way. The authors here conflate real individual-based models (i.e. where individual cells are modeled, e.g. PMID: 28531184, PMID: 22817898) with dynamic models where multiple individual species are simulated in the same environment, but there is no attempt to model individual cells (PMID: 34635859, PMID: 24742179).

Response: We agree, and we have revised this section to distinguish between these two cases (lines 171-177).

Line 194: "A major downside to these dynamical models is that they often do not scale well to many taxa or long time frames. Furthermore, large fluxes in the models will require very small-time steps so as not to overshoot consumption of metabolites in the extracellular environment, resulting in hundreds-to-thousands of steps within a single trajectory leading to lengthy simulations." Again, this statement is inaccurate. Dynamic FBA can handle hundreds of species, and the computing time scales linearly with the number of species. Scaling with time is also in general linear. Moreover, people have explored strategies for dynamic allocation of time steps. Also: it should be obvious, but simulations are long in proportion to the time course they simulate, which is unavoidable if one wants to learn about temporal variations of a community.

Response: Under appropriate error-controlling time stepping schemes, the number of time steps scales with the number of taxa in the model, even when minimizing for the number of required optimizations (see <https://doi.org/10.1371%2Fjournal.pcbi.1007786>). For large models (3+ genome-scale models), this still typically results in thousands of optimizations, which is somewhat costly if one is only interested in the steady state solution. Of course this is subjective, but for groups like ours who simulate thousands of models at a time for communities containing dozens-to-hundreds of taxa, a difference in compute time between 5 minutes (typical solution time for ctFBA for a single sample on a single CPU) and several hours is quite substantial. We do think that this limitation is reflected in the literature. For instance, all the publications mentioned above by the reviewer only simulate pairs of taxa, often using reduced

models with fewer reactions and metabolites to ensure computational tractability. We are aware of one example where larger communities with more than 100 genome-scale metabolic models were simulated with dynamic FBA (<https://doi.org/10.1101/2022.12.19.520975>), but this was limited to only two samples and it took a significant amount of computational time. Thus, we think there is a need for methods that can model hundreds of samples, containing 100+ taxa each, over a reasonable timeframe.

270: "All of the community-scale metabolic modeling strategies discussed in this perspective will ultimately require experimental validation to assess their usefulness and translational potential." I agree. However, it is surprising that there is no mention at all of the several previous papers that compared model predictions with experiments, including references mentioned in comments about Line 177 and many more.

Response: We have now included an additional figure (Fig. 2E, lines 407-413), along with additional text and references (lines 255-277), to provide a more thorough discussion of this point.

288: "Finally, we argue for the direct experimental validation of quantitative flux predictions either by isotopic labeling or by longitudinal metabolomics sampling from microbial communities (9)."

Who wouldn't argue in favor of this? The problem, as I hinted at the beginning, is that it is extraordinarily difficult to measure fluxes in a community (except for overall net fluxes in and out of the whole system). The few examples I know where measurement of fluxes across species has been attempted (for two species) is through extremely sophisticated machines like nanoSIMS (PMID: 27419912, PMID: 32788385). If the authors are aware of other promising technologies or efforts, they should mention them. So, overall I find this concluding statement quite naive, disconnected from reality (again no references, except a global metabolomics one) and with no hint to how this may be possible. At the same time, some of the other major limitations in the field of ecosystem-level modeling (including the challenges of building reliable models for newly sequenced organisms, the lack of information on biomass composition, metabolic composition of the environment, non-metabolic interactions, e.g. signaling, antibiotic killing etc.) are not mentioned at all.

Response: We agree, and we apologize for not making our argument clearer. We specifically argue for the validation of community-wide secretion and uptake rates as a scalable and simple approach. While these methods cannot validate every single flux within the system they can still provide a stringent validation of the overall model. In a similar vein, even with techniques like isotope labeling or NanoSims, one cannot quantify all possible fluxes but rather a subset of targeted fluxes. We have now made these points more clear in the text (lines 273-277).

February 27, 2023

Prof. Sean M. Gibbons
Institute for Systems Biology
401 Terry Ave. N
Room 359
Seattle, WA 98109

Re: mSystems01270-22R1 (More is different: metabolic modeling of diverse microbial communities)

Dear Prof. Gibbons:

Your manuscript has been accepted, and I am forwarding it to the ASM Journals Department for publication. For your reference, ASM Journals' address is given below. Before it can be scheduled for publication, your manuscript will be checked by the mSystems production staff to make sure that all elements meet the technical requirements for publication. They will contact you if anything needs to be revised before copyediting and production can begin. Otherwise, you will be notified when your proofs are ready to be viewed.

If you would like to submit a potential Featured Image, please email a file and a short legend to msystems@asmusa.org. Please note that we can only consider images that (i) the authors created or own and (ii) have not been previously published. By submitting, you agree that the image can be used under the same terms as the published article. File requirements: square dimensions (4" x 4"), 300 dpi resolution, RGB colorspace, TIF file format.

We recognize that the video files can become quite large, and so to avoid quality loss ASM suggests sending the video file via <https://www.wetransfer.com/>. When you have a final version of the video and the still ready to share, please send it to mSystems staff at msystems@asmusa.org.

Sincerely,

Jack Gilbert
Editor, mSystems

Journals Department
E-mail: mSystems@asmusa.org